# Anodic Catalyst Support via Titanium Dioxide-Graphene Aerogel (TiO$_2$-GA) for A Direct Methanol Fuel Cell: Response Surface Approach

**Siti Hasanah Osman** [1], **Siti Kartom Kamarudin** [1,2,*], **Sahriah Basri** [1] and **Nabila A. Karim** [1]

1 Fuel Cell Institute, Universiti Kebangsaan Malaysia, Bangi 43600, Selangor, Malaysia
2 Department of Chemical and Process Engineering, Faculty of Engineering and Built Environment, Universiti Kebangsaan Malaysia, Bangi 43600, Selangor, Malaysia
* Correspondence: ctie@ukm.edu.my

**Abstract:** The direct methanol fuel cell (DMFC) has the potential for portable applications. However, it has some drawbacks that make commercialisation difficult owing to its poor kinetic oxidation efficiency and non-economic cost. To enhance the performance of direct methanol fuel cells, various aspects should be explored, and operational parameters must be tuned. This research was carried out using an experimental setup that generated the best results to evaluate the effectiveness of these variables on electrocatalysis performance in a fuel cell system. Titanium dioxide-graphene aerogel (TiO$_2$-GA) has not yet been applied to the electrocatalysis area for fuel cell application. As a consequence, this research is an attempt to boost the effectiveness of direct methanol fuel cell electrocatalysts by incorporating bifunctional PtRu and TiO$_2$-GA. The response surface methodology (RSM) was used to regulate the best combination of operational parameters, which include the temperature of composite TiO$_2$-GA, the ratio of Pt to Ru (Pt:Ru), and the PtRu catalyst composition (wt%) as factors (input) and the current density (output) as a response for the optimisation investigation. The mass activity is determined using cyclic voltammetry (CV). The best-operating conditions were determined by RSM-based performance tests at a composition temperature of 202 °C, a Pt/Ru ratio of (1.1:1), and a catalyst composition of 22%. The best response is expected to be 564.87 mA/mg$_{PtRu}$. The verification test is performed, and the average current density is found to be 568.15 mA/mg$_{PtRu}$. It is observed that, after optimisation, the PtRu/TiO$_2$-GA had a 7.1 times higher current density as compared to commercial PtRu. As a result, a titanium dioxide-graphene aerogel has potential as an anode electrocatalyst in direct methanol fuel cells.

**Keywords:** titanium dioxide-graphene aerogel; methanol oxidation; anodic electrocatalyst; response surface methodology

## 1. Introduction

The interaction of the fuel with methanol generates energy, which is used to power an electrochemical cell, i.e., a direct methanol fuel cell, thus directly converting electrical energy without using combustion. Consequently, DMFC is an innovative application that provides facilities and resources to power portable technologies such as digital electronic equipment, notebook computers [1,2], mobile phones, and other popular devices among the general public [3,4]. It has several advantages, such as a modest system design, the possibility of using instant refueling [5], and low-volume and lightweight packaging. A DMFC could produce a significant amount of specific energy [5–7] using methanol with a low operating temperature, a high energy density, and an easy start-up [8]. However, apart from the technical advantages, DMFC has to find the best ways to eliminate There is no requirement for fuel reformation, yet it is classified as a zero-emission power system [9,10]. Nevertheless, its application is hindered by a few significant barriers that keep it from

being commercialised: high material charges [11], low efficiency, methanol crossover from the anode to the cathode [12], and catalyst poisoning during operation [9].

Generally, transition metal oxides, such as $TiO_2$ [13], $WO_3$ [14], and $SnO_2$ [15], have been revealed as carbon support alternatives for electrocatalyst stability and activity enhancement in direct fuel methanol [16]. $TiO_2$ is one of these metal oxides that researchers are particularly interested in because of its cost-effectiveness and corrosion resistance. However, the low electrical conductivity of $TiO_2$ limits its use in fuel cells. $TiO_2$ is required to overcome low electrical conductivity in support of fuel cells, which coincides with challenges in the form of catalyst structure. Furthermore, $TiO_2$ can improve MOR performance for two reasons. The first is that $TiO_2$ empowers methanol oxidation via photopic excitation due to its 3.2 eV band gap [17], and the second is that it can reduce the CO toxicity of the catalyst via its cleaning ability, by which ordinarily metal oxides like $TiO_2$ adsorb OH and, as a result, convert the catalyst-adsorbed CO to $CO_2$ to restore the active sites of Pt NPs for continuous electrocatalytic activity and effectively mitigate the CO poisoning of Pt. Shahid et al. [18] reported that $TiO_2$/Pt significantly enhanced the electrocatalytic performance of methanol oxidation. Furthermore, Zhao and team [19] reported that a Pt/graphene-$TiO_2$ catalyst with $TiO_2$ and graphene as the mixed support exhibits high activity in comparison with Pt/graphene prepared by the same process. Other than that, Lou et al. [20] identified a three-component Pt/$TiO_2$-rGO electrocatalyst with significantly improved methanol oxidation electrocatalytic performance. Interestingly, a more convenient, simple, and quick method of preparing catalysts is required. In addition to the structural design, the rate capability can be enhanced with highly conductive materials, such as $TiO_2$, for which the use of this combination has been widely adopted. Highly conductive materials include metal oxides [21], carbon [22,23], and carbon-based materials (e.g., graphene [24,25], nanofibers [26,27], and carbon nanotubes [28,29]). Graphene has had a positive effect on $TiO_2$ due to its superior conductivity, chemical stability, and high specific surface area. However, three-dimensional (3D) graphene aerogel (GA) can effectively inhibit graphene rearrangement and provide graphene-based composites with a large specific surface area, fast electron transport kinetics, and more active sites.

A substantial amount of research is indeed required before this technology can be widely used in the years ahead. Remarkable progress has been made in the essential components, such as the system, membrane, and catalyst. The majority of the use of a multi-component catalyst has been the focus of catalyst investigation. Studies have demonstrated that when Pt and Ru's catalytic activity is compared on $TiO_2$ and graphene supports, both metals show comparable activity towards methanol oxidation on both supports. The size and distribution of the metal nanoparticles on the support, the support material's surface structure, and the reaction circumstances are only a few examples of the variables that might affect the performance of the catalysts [18,30]. Basri et al. [30] recommended PtRuFeNi/MWCNT, a novel multi-component anode catalyst. Kim et al. [31] proposed the PtRu/C-Au/$TiO_2$ electrocatalyst. Abdullah et al. studied PtRu/$TiO_2$-CNF [32] in 2018 and PtRu/Mxene [33] in 2020. All studies resulted in improved reaction kinetics and DMFC performance. The membrane was also used in several boosting studies: Thiam et al. [12] established a Nafion/Pd-$SiO_2$ composite membrane; Ahmad et al. [34] introduced a Nafion-PBI-ZP hybrid membrane; Shaari and Kamarudin [35] proposed a crosslinked sodium alginate/sulfonated graphene oxide as a polymer electrolyte membrane; and You et al. [36] studied a sulfonated polyimides/sulfonated rice husk ash (SPI/sRHA) composite membrane. All studies were created to identify the methanol crossover issue in a DMFC.

The optimisation process is critical to improving the performance of DMFCs. Previously, optimisation was done using the one-factor-at-a-time method. Conversely, this approach did not take into account factor interaction and did not reflect the actual effect of the factor on the response [37]. The method of resolving the matter using mathematically analysed issues is referred to as optimisation [38]. RSM appears to be the most reassuring optimisation method that has recently been seen by researchers in a variety of fields [39–43]. This method optimises, establishes, and improves processes by combining mathematical

and statistical techniques [44]. RSM could indeed investigate the effect of independent variables in a system, either independently or in groups, and it can also reduce the number of experimental tests requisite to statistically analyse the process due to various factors [45]. The issue that our assessment is attempting to address is the 'optimal' value of the factors influencing electrocatalytic activity for methanol oxidation. These investigations discovered that the mathematical model of RSM may be employed for exact estimates and optimisation. Although DMFC optimisation has been studied extensively, electrocatalyst optimisation has received little attention.

Throughout this investigation, the bifunctional catalyst PtRu combined with $TiO_2$-GA is developed for the first time for use in DMFCs. As a consequence, the purpose of this study is to develop and optimise the $TiO_2$-GA integrated electrocatalyst performance for MOR using the RSM idea. To gain a better understanding of RSM, three parameters are manipulated: composite temperature, Pt/Ru ratio, and PtRu catalyst composition as well as the response of the current density to electrocatalyst activity in DMFC performance. Following the $TiO_2$-GA synthesis two methods are combined to make a fine composite, namely the hydrothermal and freeze-drying methods. PtRu is then coated onto $TiO_2$-GA nanoparticles. An X-ray diffraction analysis (XRD), Brunauer-Emmett-Teller (BET), field emission scanning electron microscope (FESEM), and transmission electron microscopy (TEM) were used to examine the PtRu/$TiO_2$-GA electrocatalyst for physical. The study is made more fascinating by the use of cyclic voltammetry to examine the RSM optimiser depending on three manipulation parameters and one response. The RSM was used to create a model that corresponded to the parameters analysed, and the findings of that analysis could be used in subsequent design space analyses. This research has become more intriguing with the demonstration that MOR performance is superior to commercial DMFC electrocatalysts in which the strong bond between the PtRu of the catalyst itself and the supporter, i.e., $TiO_2$-GA, in a unique 3D structure encompasses a broad surface-active site on the electrocatalyst to react.

## 2. Results and Discussion

### 2.1. Structure of Synthesised PtRu/TiO_2-GA Electrocatalyst and Electrochemical Testing

Throughout this research, FESEM was utilised to investigate the morphology of the bifunctional Pt and Ru distributions on the $TiO_2$-GA structure as well as the elemental and external mapping of the electrocatalyst. According to this, it is advantageous to boost the activity of the catalyst during the CV electrochemical test. Figure S1 (supplementary document) depicts the $TiO_2$-GA and electrocatalyst morphologies. $TiO_2$-GA and PtRu/$TiO_2$-GA FESEM images are acquired at magnifications of 500X and 5kX, respectively. Figure 1a of $TiO_2$-GA shows that the 3D $TiO_2$-GA structure was successfully developed, with a better-defined porosity structure within the ultrathin layer of the aerogel matrix. Figure 1b is the FESEM image of the PtRu/$TiO_2$-GA electrocatalyst, showing large particles dispersed and covering the 3D $TiO_2$-GA structure. Instead of that, Figure 1c described the original figure of the EDX and mapping for overall elements' particles. EDX and mapping analysis are used to further determine the existence of particles in $TiO_2$-GA, as shown in Figure 1d–h. The findings demonstrate that five elements, namely Pt, Ru, Ti, C, and O, are present in the electrocatalyst. Most of these elements are required in the electrocatalyst, and no impurities are present in the sample. Pt and Ru particles are similarly dispersed over the $TiO_2$-GA structure, according to the electrocatalyst mapping study. It would be useful to establish active response regions throughout the catalytic activity, potentially improving MOR. Nevertheless, there are a few accumulations of Pt and Ru on the sample as a result of the influence of NaOH overuse during the adjusting of the pH in the deposition process [46].

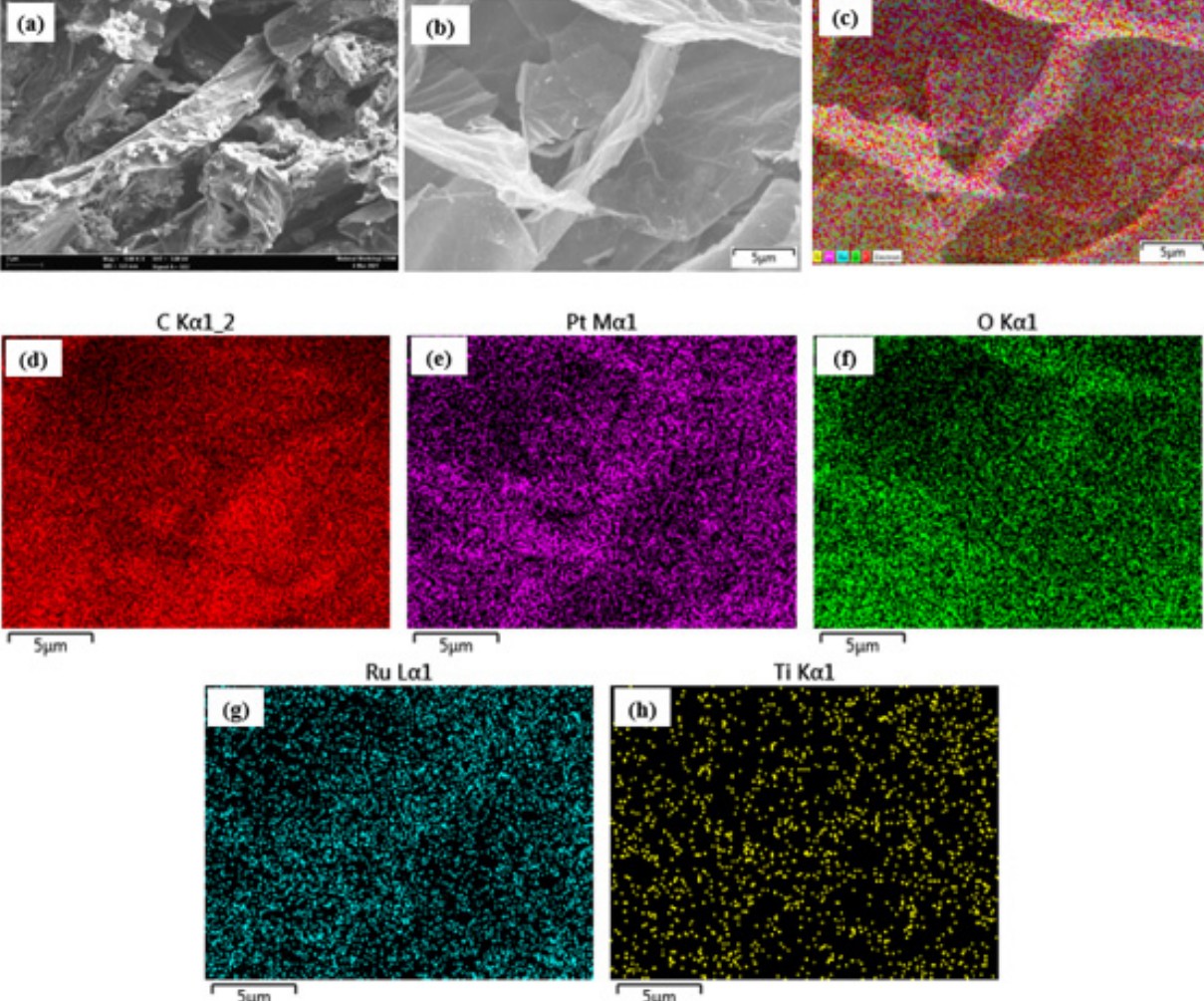

**Figure 1.** Surface morphology for (**a**) SEM of TiO$_2$-GA, (**b**) SEM of PtRu/TiO$_2$-GA, (**c**) EDX analysis of PtRu/TiO$_2$-GA, and (**d**–**h**) Mapping analysis for PtRu/TiO$_2$-GA.

The "electron relay effect" is a mechanism through which the support material (carbon) can deliver electrons to the Pt catalyst. During a sequence of electron transfers, the carbon support material serves as a source of electrons, which are then transported to the Pt catalyst. Several stages can be taken to move the electrons from the carbon support material to the Pt catalyst:

1. Whenever the electrons from the carbon support material interact with the Pt catalyst, they are first moved to the Pt-C bond contact.
2. After being transmitted from the Pt-C interface to the Pt surface, these electrons can subsequently be employed to speed up the methanol oxidation process.
3. Once the Pt catalyst uses these electrons to speed up the process, it may eventually lose its effectiveness. Yet, by supplying a source of electrons, the electron transfer from the support material contributes to maintaining the catalytic activity of the Pt catalyst.

In general, this process of electron transfer from the support to the catalyst promotes the MOR by offering a source of electrons that the Pt catalyst can exploit to speed up the reaction. This can increase the overall effectiveness of the reaction and enable the catalyst to sustain its activity over time.

Furthermore, the bifunctional mechanism that shifts the onset potential to a low potential region also improves catalytic performance during the MOR process. Other second metals, such as Ru, lower the potential region and increase catalytic activity, which is consistent with the bifunctional mechanism.

The electro-oxidation of methanol using a PtRu catalyst can be determined by the following equations, depending on the bifunctional mechanism:

$$Pt + CH_3OH \rightarrow Pt - CH_3OH_{ads} \rightarrow Pt - COH_{ads} \rightarrow 3H + 3e^- \rightarrow Pt - CO_{ads} + H^+ + e-, \quad (1)$$

$$Ru + H_2O \rightarrow Ru - OH_{ads} + H^+ + e^-, \quad (2)$$

$$Pt - COH_{ads} + Ru \text{ - } Oh_{ads} \rightarrow Pt + Ru + CO_2 + 2H^+ + 2e^-, \quad (3)$$

$$Pt - CO_{ads} + Ru - OH_{ads} \rightarrow Pt + Ru + CO_2 + H^+ + e^- \quad (4)$$

The proposed mechanism of PtRu/TiO$_2$-GA for the enhanced catalytic MOR activity as shown in Figure 2 can be defined as follows: (1) The unusual electronic reaction of the metal support between PtRu and TiO$_2$ results in electron transfer from the support to the metal, resulting in an increase in electron density at PtRu downwards, corresponding to the PtRu d-band centre. In general, the centre of the d-band is driven by the ability of surface d-electrons to participate in adsorption bonding. Based on the downward displacement of the d-band centre, reducing the metal back bond to CO can reduce the bonding energy between CO and PtRu atoms [47–49]. (2) In the catalyst, OH$_{ads,}$ or oxygen-containing species, can be adsorbed. TiO$_2$ has the unique ability to convert CO-poisoning species (CO$_{ads}$) on PtRu bifunctional to CO$_2$, releasing the active sites of PtRu bifunctional by promoting electrooxidation activity towards methanol and CO [50]. (3) Pt, Ru, graphene aerogel, and TiO$_2$ have a good synergy effect, resulting in a four-junction structure that effectively improves electrocatalytic performance [20].

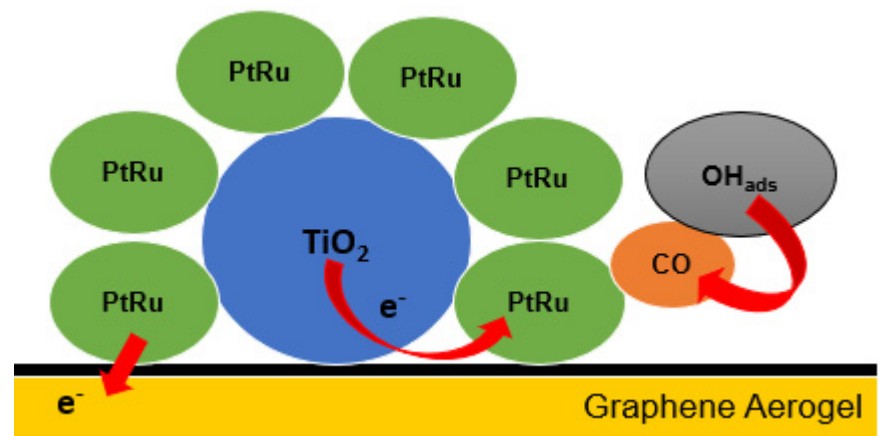

**Figure 2.** The proposed PtRu/TiO$_2$−GA pathway for enhanced catalytic MOR activity.

### 2.2. One-Factor-At-A-Time

The anode electrocatalyst, mainly composed of PtRu/TiO$_2$-GA for a DMFC, is the first time it has been accomplished in this investigation. This electrocatalyst is made up of four major components: the primary catalysts are Pt and Ru, with TiO$_2$ and graphene performing as support catalysts in an aerogel framework. Pt is widely regarded as the most effective catalyst for reactions where the carbon monoxide (CO) bond intensity can be reduced using reactions, such as hydrogen oxidation, oxygen reduction, and a combination of Pt and Ru, thus boosting catalytic performance. The inclusion of TiO$_2$ provides high thermal and electrochemical stability, whereas the electrocatalyst reaction surface area is increased by using an aerogel structure. As compared to conventional catalysts, all of these components work together to dramatically improve performance. Nevertheless, modifications of several of the elements might have an impact on the total performance of the catalyst, so identifying the optimal quantities of the catalysts is crucial to enhancing performance. The composition of this material versus TiO$_2$-GA must be developed to

minimise production costs because the PtRu catalyst is a critical component that comes at a significant expense.

There are three parameters selected by respondents in this investigation, of which the major parameter is the temperature of the composition, °C (TiO$_2$-GA), the ratio of Pt to Ru, and catalyst composition interaction; PtRu is the main catalyst, while TiO$_2$-GA is the supporting catalyst. Most variables measured are optimised for half-cell performance, and the parameters were preferred for this study because they have a powerful influence on the DMFC performance based on electrocatalyst, which is comparable to what other scholars have found [22,32,51–54]. This value has been chosen based on the level of investigation from prior studies, which is a mix of PtRu and metal oxide support and graphene aerogel as an electrocatalyst [22,32,51–54]. The variety selected in the screening process was included in this research for the temperature of composition TiO$_2$-GA, which is varied in the range of 180–220 °C, the ratio of Pt to Ru in the range of 0.5–2, and the catalyst composition in the range of 10–30%.

Figure 3a–c present graphs illustrating the determined current density for the temperature of composition TiO$_2$-GA, the ratio of Pt to Ru, and catalyst composition over the experimental range. The maximum current density acquired using a 200 °C composite temperature and a 1:1 ratio is shown in the half-cell DMFC performance results (Pt:Ru). In addition, in the screening process, a catalyst composition of 20% by weight is used. The curve of the graph, on the other hand, shows that the most effective performance of DMFC can be achieved at temperature compositions of 180–220 °C. The second factor, the ratio of Pt to Ru in the range of 0.5–1.5, has a graph pattern that shows the ratio range increasing and decreasing. The catalyst composition in the range is the final factor in this study (15–25%). The results showed that the maximum current density could be achieved between 107.96 and 554.16 mA/mg$_{PtRu}$, with the greatest result from the screening process occurring at 608.17 mA/mg$_{PtRu}$. In the RSM optimisation process, the range of values for the variables is used.

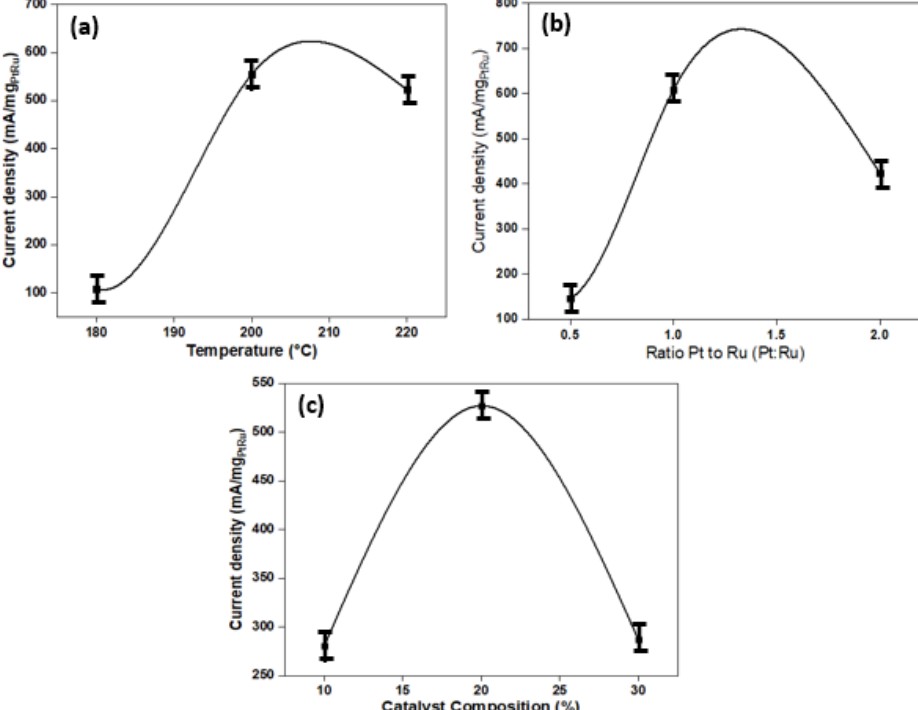

**Figure 3.** (**a**) The effect of temperature composite on the current density. (**b**) The effect of the ratio Pt to Ru (Pt:Ru) on the current density. (**c**) The effect of catalyst composition on the current density.

### 2.3. Optimisation Using RSM

The RSM CCD method optimisation process used 20 sessions for three study components. The experimental results for current density, including all assessments, are shown in Table 1. The regression technique was used to fit a quadratic approach to the results obtained. The current density response is modelled via Equation (1).

$$Y = 555.54 + 45.05 \times A + 22.64 \times B + 27.41 \times C + 4.16 \times A \times B + 1.76 \times A \times C$$
$$+ 28.35 \times B \times C + \tag{5}$$
$$(-235.21) \times A2 + (-164.13) \times B2 + (-35.57) \times C2$$

where Y denotes the current density ($mA/mg_{PtRu}$) and A, B, and C denote the temperature of the composition (°C), the ratio of Pt to Ru, and catalyst composition, respectively. ANOVA is a statistical analysis that compares modifications in variable-level combinations to adjustments due to random errors in response measurement [37]. The *p*-value and F-value are calculated using ANOVA, and a smaller *p*-value and a larger F-value indicate that interpreting design factor variability throughout mean data has more confidence.

**Table 1.** Summary results of the 20 experiments involved.

| Run | Factor A | Factor B | Factor C | Response 1 | |
|-----|----------|----------|----------|----------------|--------------|
| | | | | **Predicted Value** | **Actual Value** |
| 1 | 200.00 | 1.00 | 20.00 | 555.53 | 617.67 |
| 2 | 200.00 | 1.00 | 15.00 | 492.56 | 460.39 |
| 3 | 200.00 | 1.00 | 20.00 | 555.53 | 537.40 |
| 4 | 200.00 | 1.00 | 25.00 | 547.38 | 522.85 |
| 5 | 180.00 | 1.00 | 20.00 | 275.27 | 237.45 |
| 6 | 220.00 | 1.50 | 25.00 | 249.99 | 249.02 |
| 7 | 180.00 | 0.50 | 25.00 | 54.39 | 60.5 |
| 8 | 220.00 | 0.50 | 25.00 | 139.71 | 148.20 |
| 9 | 220.00 | 1.50 | 15.00 | 134.95 | 143.01 |
| 10 | 200.00 | 1.00 | 20.00 | 555.53 | 523.22 |
| 11 | 200.00 | 1.50 | 20.00 | 414.05 | 390.39 |
| 12 | 200.00 | 0.50 | 20.00 | 368.77 | 335.74 |
| 13 | 180.00 | 1.50 | 15.00 | 40.05 | 45.73 |
| 14 | 220.00 | 0.50 | 15.00 | 138.07 | 141.35 |
| 15 | 200.00 | 1.00 | 20.00 | 555.53 | 608.41 |
| 16 | 180.00 | 0.50 | 15.00 | 59.79 | 74.94 |
| 17 | 200.00 | 1.00 | 20.00 | 555.53 | 607.98 |
| 18 | 220.00 | 1.00 | 20.00 | 365.37 | 346.51 |
| 19 | 180.00 | 1.50 | 25.00 | 148.06 | 158.94 |
| 20 | 200.00 | 1.00 | 20.00 | 555.53 | 551.9 |

Table 2 shows the outcomes of an ANOVA for a quadratic response surface model as well as the significance of each coefficient. This model's F-value and *p*-value > F are 52.34 and 0.05, respectively, indicating its significance. There was only a 0.01% probability that the results were due to noise. Besides, the 0.90 lack of fit verifies its significance; the result demonstrates that this fitted model is relevant. These findings indicate that such a model could be used to predict the outcome of research in this area. The standard deviation and the determination coefficient, $R^2$, may be used to assess the appropriateness of this model. The standard deviation and $R^2$ for this model were 40.54 and 0.9792, respectively. This means that the model can account for 97.92% of the total variation in the reaction. In the meantime, the 'Pred $R^2$' of 0.9409 agrees reasonably with the 'Adj $R^2$' of 0.9605. The signal-to-noise ratio can be measured with greater precision, with a value greater than 4 being favoured. The adequate precision for the current density model is 17.984, indicating an adequate signal. To recap, the model accurately predicts future responses by fitting the experimental data. A plot of the residuals is included as part of the diagnostic model.

**Table 2.** Results of ANOVA analysis for current density model.

| Source | Sum of Squares | DF | Mean Square | F-Value | *p*-Value Prob > F |
|---|---|---|---|---|---|
| Model | $7.74 \times 10^5$ | 9 | 86,002.39 | 52.34 | <0.0001 significant |
| A: Temperature | 20,298.36 | 1 | 20,298.36 | 12.35 | 0.0056 |
| B: Ratio Pt to Ru (Pt:Ru) | 5124.56 | 1 | 5124.56 | 3.12 | 0.1078 |
| C: Catalyst Composition (wt%) | 7512.59 | 1 | 7512.59 | 4.57 | 0.0582 |
| $A^2$ | $1.521 \times 10^5$ | 1 | $1.521 \times 10^5$ | 92.60 | <0.0001 |
| $B^2$ | 74,077.24 | 1 | 74,077.24 | 45.08 | <0.0001 |
| $C^2$ | 3479.08 | 1 | 3479.08 | 2.12 | 0.1763 |
| AB | 138.13 | 1 | 138.13 | 0.084 | 0.7778 |
| AC | 24.79 | 1 | 24.79 | 0.015 | 0.9047 |
| BC | 6430.57 | 1 | 6430.57 | 3.91 | 0.0761 |
| Residual | 16,431.12 | 10 | 1643.11 | | |
| Lack of Fit | 7780.73 | 5 | 1556.15 | 0.90 | 0.5449 not significant |
| Pure Error | 8650.39 | 5 | 1730.08 | | |
| Correlation Total | $7.905 \times 10^5$ | 19 | | | |
| Standard Deviation | 40.54 | | $R^2$ | 0.9792 | |
| Mean | 338.08 | | Adj $R^2$ | 0.9605 | |
| | | | Pred $R^2$ | 0.9409 | |
| | | | Adeq $R^2$ | 17.984 | |

The diagnostic portion of RSM analysis is another process. In this section, graphs will be used to properly assess the model fit and transformation preference. Figure 4 depicts the model fit error, which is commonly referred to as a residual plot. Figure 4a shows a straight line for the residual normal probability plot, suggesting that the residual follows the normal distribution and contains acceptable normal error components. Figure 4b depicts a residual versus projected value plot of the model response, with a straight line at '0', implying that the expected variance for such a model is constant. Simultaneously, the recommended quadratic model for the current density model appears to be appropriate, and all plots fall between the upper and lower red lines with no discernible pattern.

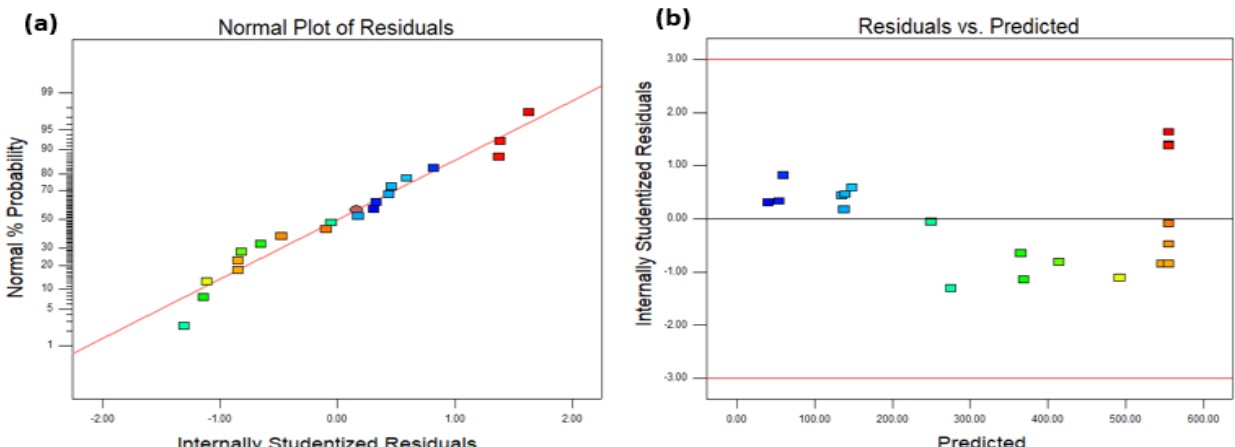

**Figure 4.** A residual plot for the current density model; (**a**) normal plot of residual; (**b**) residual vs. predicted plot.

The predicted vs. actual plot is used to identify values that are challenging for the model to predict [30], which is shown in Figure 5a. The plotted data is just in the centre of the graph and forms a 45° perpendicular line. This requires the model's ability to correctly predict the response. Figure 5b depicts a perturbation plot, which demonstrates how the factors can influence the response. As previously stated, factors A, B, and C are the temperature of the composite, the Pt/Ru ratio, and the catalyst composition, respectively. The actual values are set to the 'coded 0' midpoint for all of the factors: A: 202 °C, B: 1.1,

and C: 22 wt%. The perturbation graph is created by changing one factor at a time over the response value. Overall, three factors have a steep slope in the plot, showing that they are all influenced by or sensitive to the experimental response and are important to the system design. Nevertheless, factor A's graph has a little steeper gradient than factors B and C, showing that factor A has a larger impact on the response value.

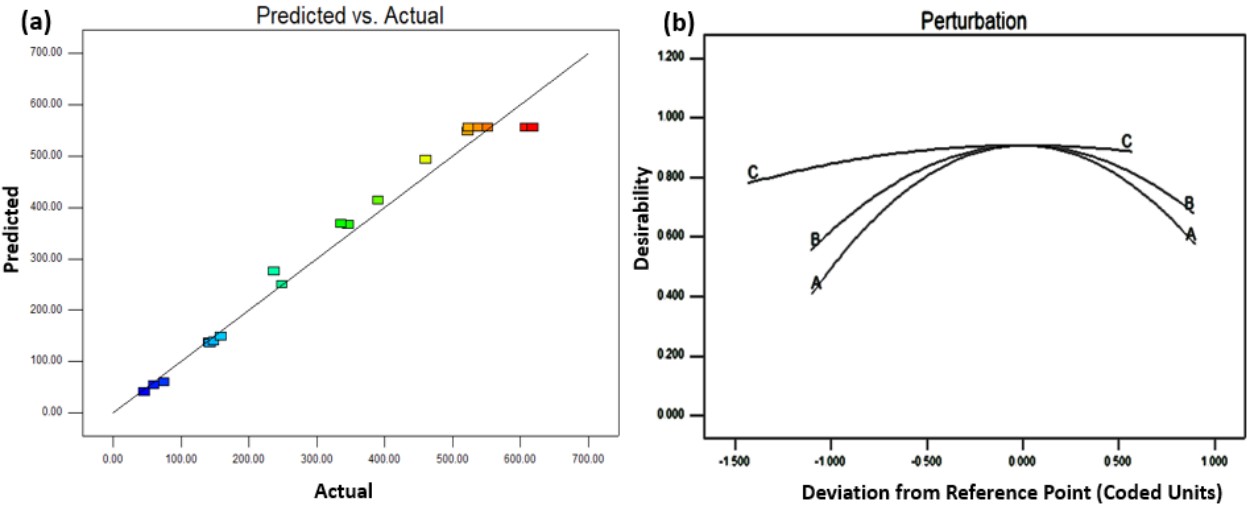

**Figure 5.** (**a**) Predicted vs. actual plot, and (**b**) perturbation plot for the current density model.

Predicting or evaluating the mean reaction is what response surface analysis includes at a specific point in the process [44]. Figure 6 shows the response surface for a contour plot in two dimensions (2D) and a surface plot in three dimensions (3D). The response surface is made up of an analysis of two factors, namely AB, AC, and BC, as well as the current density's response. Nevertheless, Figure 6 depicts an examination of current density factors A and B. The relationship between components A and B as well as the response, as depicted in the 2D contour plot, has a specific result in Figure 6a. The plot shows that increasing both factors increases the response. After a certain point, the response trends begin to decline even as the factor value increases. This is considered the optimum point, when the most response to the model may be correlated by the optimum factors. The other two components, AC and BC, have essentially comparable reaction patterns. In the red area of the contour plot, also referred to as the high response value area, the ideal location for factors can be found. The contour plot displayed the same patterns as the 3D surface plot in Figure 6b, and the distinct peak for all parameters represents an optimum point that reached the maximum response. The AB, AC, and BC factors have a similar pattern. The 3D graph predictive analysis of a second-order model, according to the relevant literature, reveals that the quadratic model is similar to the existing density model.

The following section deals with RSM optimisation evaluation. Alienated numerical optimisation, graphical optimisation, point prediction, and confirmation are the four main categories. The numerical optimisation classifications are required to determine the aspirations and estimate the factors of the optimal condition to yield the maximum response as determined by the model's goals. Following graphical optimisation, Figure 7a,b show the 2D contour plot for desirability with a response prediction value (for instance, in terms of the AB factor). This model's prediction values for desirability and response are 0.908 and 564.866 mA/mg$_{PtRu}$, respectively, as shown in the plot in the high response area. Figure 7 displays the point predictions for each of the model's optimum factors. The graph shows that when two graphs with significant desirability overlap, those three factors achieve an ideal position.

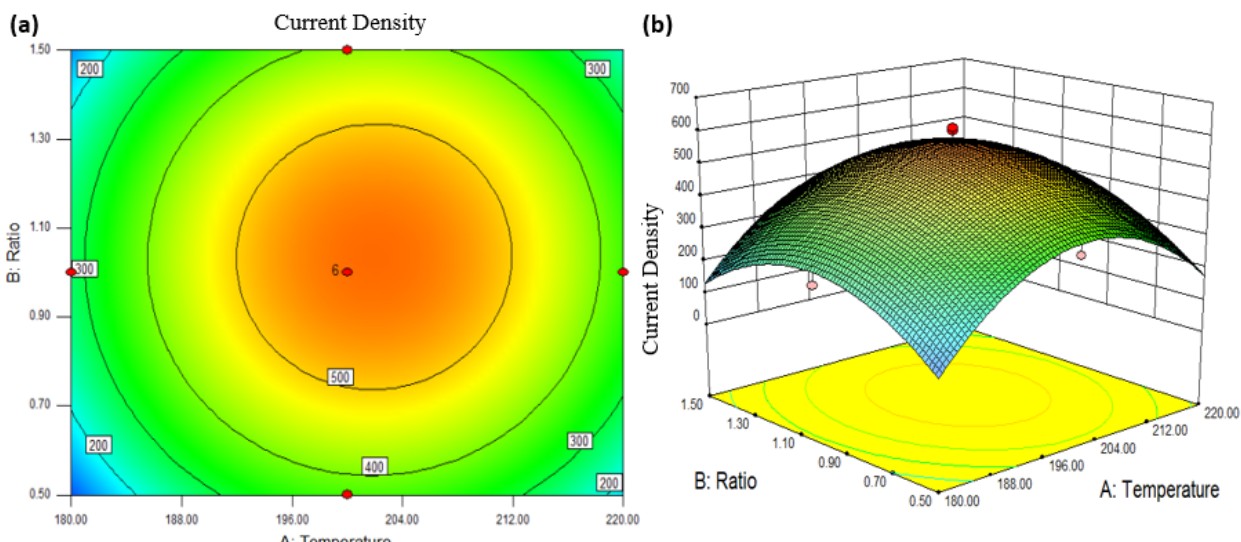

**Figure 6.** Response surface between factors; Pt:Ru ratio and a composite temperature, with the response; current density, (**a**) 2D contour, and (**b**) 3D surface plot.

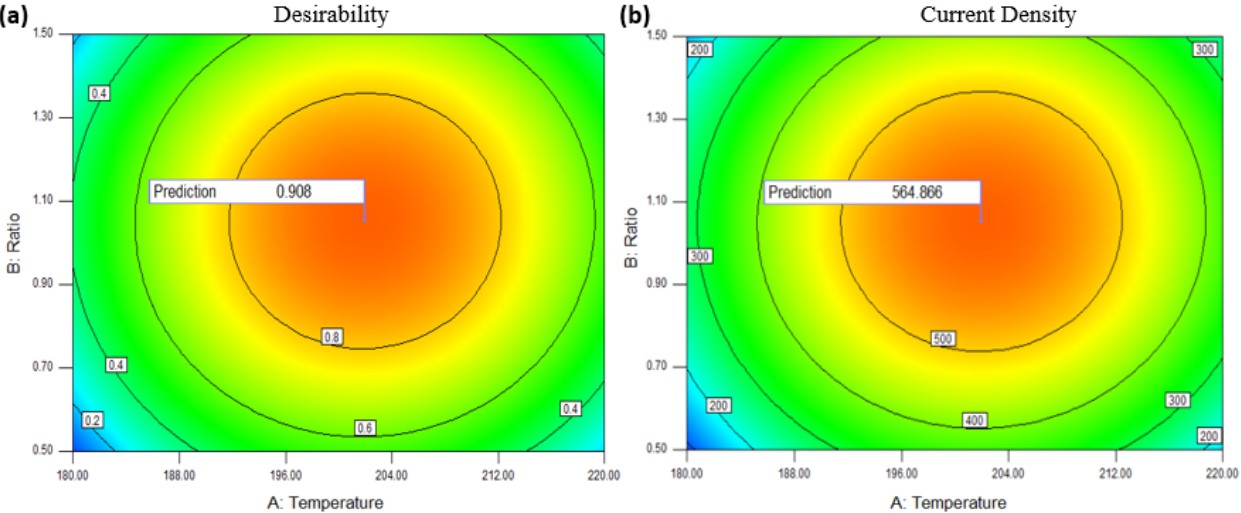

**Figure 7.** 2D contour plots for, (**a**) Desirability and (**b**) Current density in terms of AB factors.

The RSM also examined the optimum factors: A (temperature composite): 202 °C, B (ratio Pt/Ru): 1.1 and C (catalyst composition): 22 wt%. Validation is then used to compare the model's predicted results to the experimental results. To get the average, the validation test with the value of the optimal factor is carried out in triplicate, and the outcome is shown in Table 3.

**Table 3.** Validation of current density model.

| Factor A (°C) | Factor B | Factor C (wt%) | Mass Activity (mA/mg$_{PtRu}$) | | | | | Error (%) |
|---|---|---|---|---|---|---|---|---|
| | | | Prediction | 1 | 2 | 3 | Average | |
| 202 | 1.1 | 22 | 564.87 | 529.40 | 608.17 | 566.89 | 568.15 | 0.6 |

Figure 8 depicts the current density graph from the validation test. The CV test typically provides electrochemical measurements and is used to calculate the response value of this model. Once compared to the RSM anticipated value, the average validation test

result was 568.15 mA/mg PtRu, which correlated to a peak potential of 0.65 V vs. Ag/AgCl with just a 0.6% error. The small inaccuracy shows that with the optimal temperature composite, Pt/Ru ratio, and catalyst composition, the best current density response may be achieved. This situation also guarantees that the RSM analysis model is effective and applicable.

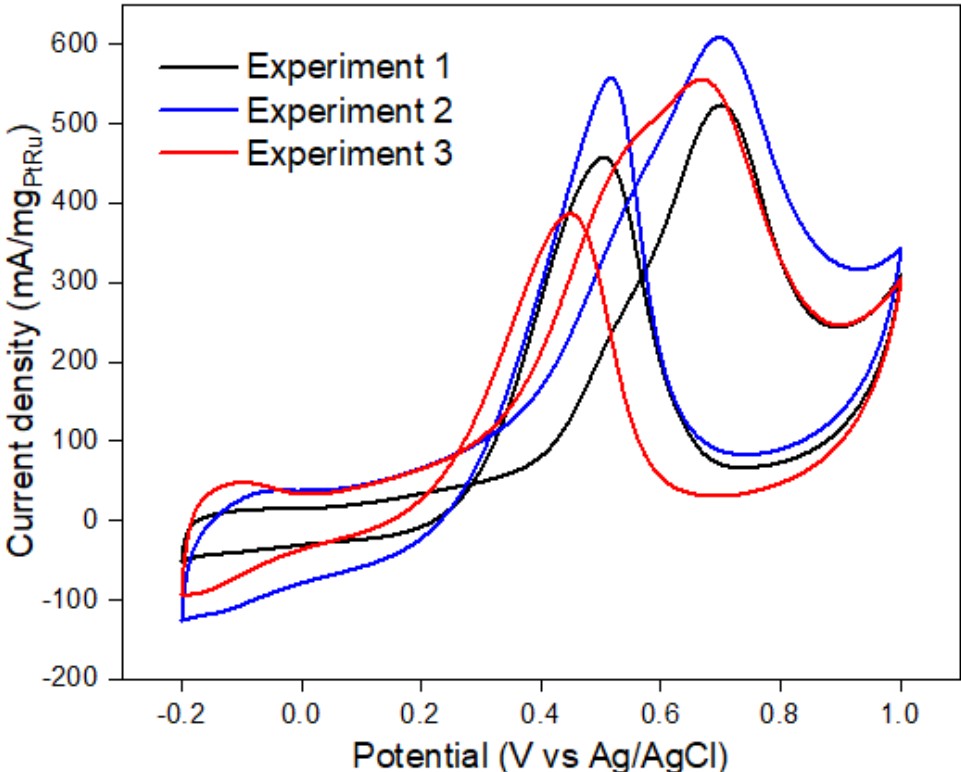

**Figure 8.** Experiment to validate the present current density model.

*2.4. Interactions of Synthesis PtRu/TiO$_2$-GA Optimisation*

2.4.1. Physical Characterisation

Pattern and crystal structure were used to investigate the physical characteristics of the PtRu/TiO$_2$-GA electrocatalyst before and after optimisation compared to PtRu/C. This investigation was carried out utilising an x-ray diffractometer (XRD) in the 5–80° range with 2θ resolution. The PtRu/TiO$_2$-GA electrocatalyst XRD pattern revealed a diffraction peak for all of the included Pt, Ru, TiO$_2$, and C. As a control experiment, PtRu/C was used. The diffraction peak (2θ = 40.3°) does not appear clearly on other electrocatalyst samples after the reduction procedure. Furthermore, no discernible difference is detected in the XRD patterns of PtRu/TiO$_2$-GA and PtRu/TiO$_2$-GA$_{Opt}$. The other peaks are in accordance with the anatase phase of the TiO$_2$ standard spectrum (JCPDS No. 21-1272). TiO$_2$ anatase phase has greater diffraction peaks at 25.4° (1 0 1), 37.7° (2 0 0), 53.8° (2 1 1), 55° (2 0 4), and 63° (1 1 6) than TiO$_2$ rutile, which has 27° (1 1 0), 35.6° (1 0 1), and 39.5° (1 1 1).

Diffraction peaks at planes (1 1 1), (2 0 0), and (2 2 0) at about 2θ are 40.7°, 47.3°, and 67.0°, respectively, which detected the Pt nanoparticle, which is compatible with the face-centred cubic (fcc) structure of Pt [55]. while the diffraction peaks for Ru are in the same plane, namely (1 1 1), (2 0 0), (2 0 0), and (2 2 0). Due to the strong crystallinity of Pt-Ru nanoparticles in the electrocatalyst system, low diffraction peaks of TiO$_2$-GA are difficult to identify following the dispersion of Pt-Ru particles on the TiO$_2$-GA support. The creation of a bimetal or Pt alloy in each synthesised electrocatalyst sample is also indicated by the yield of a single diffraction peak shared by PtRu in each of these electrocatalyst diffractograms [56,57].

The Bragg angle is evident in the range of 25–60° for all electrocatalyst samples, as shown in Figure 9. This suggests that the catalyst has a bimetallic or alloy interaction [58]. Meanwhile, when comparing PtRu/C to PtRu/TiO$_2$-GA before and after optimization, weak and broad peak intensities were detected. Regarding the crystal size values reflecting the high dispersion in the produced sample, no substantial change in peak intensity can be noticed in this situation. Table 3 lists the crystal size values available using Eva software for analysing XRD results.

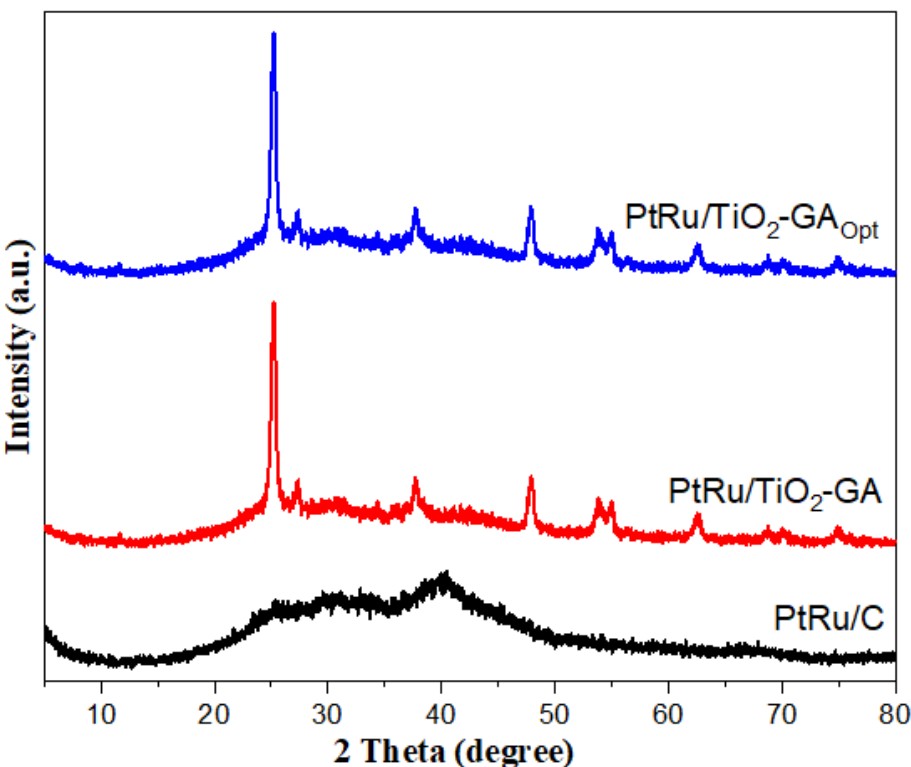

**Figure 9.** XRD patterns for prepared samples.

Additionally, N$_2$ adsorption-desorption isotherms and pore size distribution analyses were performed at a temperature of 77 K to investigate the porous structure and specific surface area of TiO$_2$-GA composites.

The N$_2$ adsorption/desorption plots (Figure 10) show that PtRu/TiO$_2$-GA$_{Opt}$ has a significantly larger surface area than PtRu/TiO$_2$-GA and PtRu/C, as shown in Table 4. The PtRu/TiO$_2$-GA$_{Opt}$ electrocatalyst had the lowest BET surface area of 50.59 m$^2$/g, followed by the PtRu/TiO$_2$-GA and PtRu/C electrocatalysts in that order. The findings of this study are comparable to those of those who found that the PtRu/C electrocatalyst had a significantly greater surface area than the metal oxide composite electrocatalyst. According to the BET adsorption/desorption isothermal curves in Figure 10, all produced electrocatalyst materials had type IV isothermal curves (based on IUPAC classification) with a little indentation of the observed width indicating H3 hysteria [59,60].

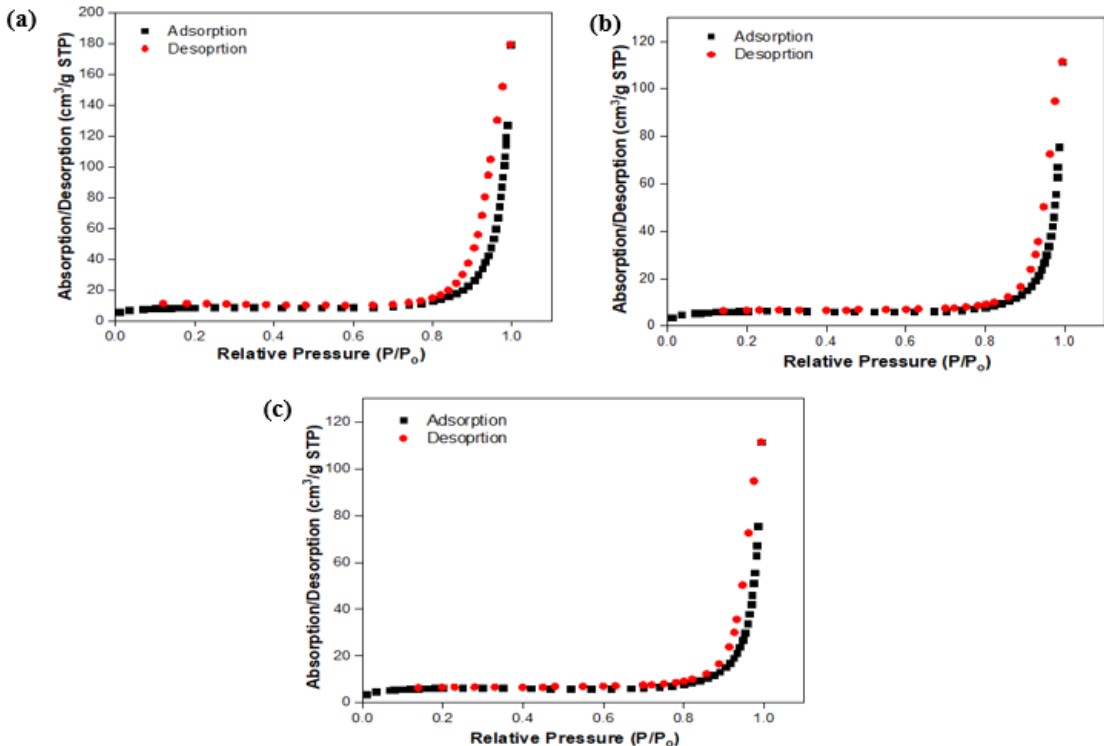

**Figure 10.** Adsorption/desorption linear isotherm: (**a**) PtRu/C; (**b**) PtRu/TiO$_2$-GA; and (**c**) PtRu/TiO$_2$-GA$_{Opt}$.

**Table 4.** BET analysis results for size and porosity properties of PtRu/TiO$_2$-GA$_{Opt}$, PtRu/TiO$_2$-GA, and PtRu/C electrocatalysts.

| Electrocatalyst | $S_{BET}$ (m$^2$ g$^{-1}$) | $V_{Total\ Pore}$ | DPore (nm) | Types of Pore |
|---|---|---|---|---|
| PtRu/TiO$_2$-GA$_{Opt}$ | 19.30 | 0.13 | 24.22 | Meso |
| PtRu/TiO$_2$-GA | 22.36 | 0.45 | 31.05 | Meso |
| PtRu/C | 31.15 | 0.61 | 32.63 | Meso |

Furthermore, at the relative pressure position P/Po > 0.6, the desorption branch was greater than the adsorption branch. While the porosity structure of the electrocatalysts presented is a mesopore type with an average diameter range of 2–50 nm, this can be attributed in large part to the huge gaps seen in the electrocatalyst lattice [61,62]. Additionally, the observations on the nitrogen isotherm adsorption/dehydration graph appear flat at low pressures, i.e., P/Po $\leq$ 0.6, which could be related to microbe adsorption in the sample. The adsorption of monolayer and/or multilayer nitrogen molecules on the mesostructure increases the sample's adsorption capacity at relatively high-pressure areas (0.6 $\leq$ P/Po $\leq$ 1.0) while catalysts spread on the support, which can boost the catalytic activity's stability and performance. The BET pore diameter findings are identical to the XRD particle size analysis results. The smaller the particle, the higher the surface-to-volume ratio, which might result in high surface reactivity and solubility. This notion is supported by the current study, which found that PtRu/TiO$_2$-GA$_{Opt}$ electrocatalysts have smaller pore widths and exhibit stronger MOR activity than PtRu/TiO$_2$-GA and PtRu/C electrocatalysts.

The degree of homogeneity of the distribution of each sample can be identified using FESEM mapping pictures. The interconnected 3D porous networks with open pores ranging in size from a few hundred nanometers to tens of micrometres are visible for both PtRu/TiO$_2$-GA and PtRu/TiO$_2$-GA$_{Opt}$ composites. In the ultra-thin layer of the aerogel matrix, the PtRu/TiO$_2$-GA$_{Opt}$ electrocatalyst has a more prominent porosity structure.

Figure S2a (supplementary document) shows a nanocomposite composed of randomly aggregated, thin, crumpled graphene sheets that are intimately linked. Furthermore, it demonstrates that a nanocomposite composed of randomly aggregated, wrinkling graphene sheets is tightly related to one another. Figure S2b (supplementary document) depicts the distribution of $TiO_2$ and Pt nanoparticles over curly graphene sheets. Furthermore, in FESEM mapping, the uniformity of PtRu/$TiO_2$-GA$_{Opt}$ is higher than that of PtRu/$TiO_2$-GA, which can be related to the uniformity of the PtRu catalyst resulting from its reporter when the reduction process happens. EDX analysis was also performed to further analyse the qualitative elemental analysis of the inserted composite; see Figure S2c,d (supplementary document). The sample contains Ti, Pt, Ru, O, and C, but no other elemental contaminants have been discovered. Based on FESEM and TEM images, i.e., Figure S2 (supplementary document) and Figure S3, no pores are visible on the catalyst nanoparticles, which is consistent with the low BET surface area study because this substance is a metallic element with high porosity. However, the surface area of the BET does not provide an overall view of the surface of the electrocatalyst's active site. As a result, electrochemical analysis can provide precise information about the active surface area of the catalyst.

The TEM images of PtRu/$TiO_2$-GA before and after optimisation to determine the morphology of all samples created using the microwave-assisted alcohol reduction process are shown in the figure. The image clearly shows the spherical shape of the carbon support material. This spherical carbon could be associated with graphene sheets to some extent, helping to increase the pores of the aerogel by cooperative dispersion, resulting in increased porosity [63]. Both samples have an estimated diameter size of 2–5 nm, allowing the samples to be classified in nano size parallel to the XRD crystal particle size as given in Table 5. When comparing the PtRu/$TiO_2$-GA sample area to the PtRu/$TiO_2$-GA$_{Opt}$ sample area, TEM examination revealed only tiny clumps. This situation indirectly affects the electrocatalyst's maximum function in the methanol oxidation reaction (MOR). Van der Waals bonding in the particles may cause agglomeration. In conclusion, the reduction approach for synthesising PtRu/$TiO_2$-GA using ethylene glycol solvent and microwave technology was successful in a short time and was effective in the manufacture of dual-function nanoparticles on $TiO_2$-GA support without isolating the support.

**Table 5.** The porosity properties of samples.

| Properties Sample Name | $S_{BET}$ (m$^2$ g$^{-1}$) | $V_{Total\ pore}$ | XRD (Crystallite Size (nm)) | Types of Pores |
|---|---|---|---|---|
| PtRu/C | 26.11 | 0.17 | 3.5 | Meso |
| PtRu/$TiO_2$-GA | 19.30 | 0.11 | 3.1 | Meso |
| PtRu/$TiO_2$-GA$_{Opt}$ | 17.15 | 0.10 | 2.5 | Meso |

2.4.2. Electrochemical Evolution

The most important aspects of CV and CA are needed in electrochemical testing. CV is necessary to measure electrocatalytic performance in this part, whereas CA is required to test electrocatalytic durability and stability. Figure 11a depicts the CV profiles of all catalysts in a 0.5 M $H_2SO_4$ solution at potentials ranging from $-0.2$ to 1.0 V. ECSA could be used to calculate the surface area of PtRu nanoparticles in electrocatalysts [64]. This happens when there is limited adsorption at the activation site, requiring electrode current cycles within a voltage range. The total charge required for monolayer adsorption and desorption identifies the reactive surface site for ECSA in the range of $-0.2$ to 0.1 V [65]. Table 6 contains the ECSA results, and CV measurements were used to evaluate ECSA using the equation below:

$$\text{ECSA}\left(\text{m}^2\text{g}_{Pt}^{-1}\right) = \frac{Q}{\Gamma \cdot W_{Pt}}$$

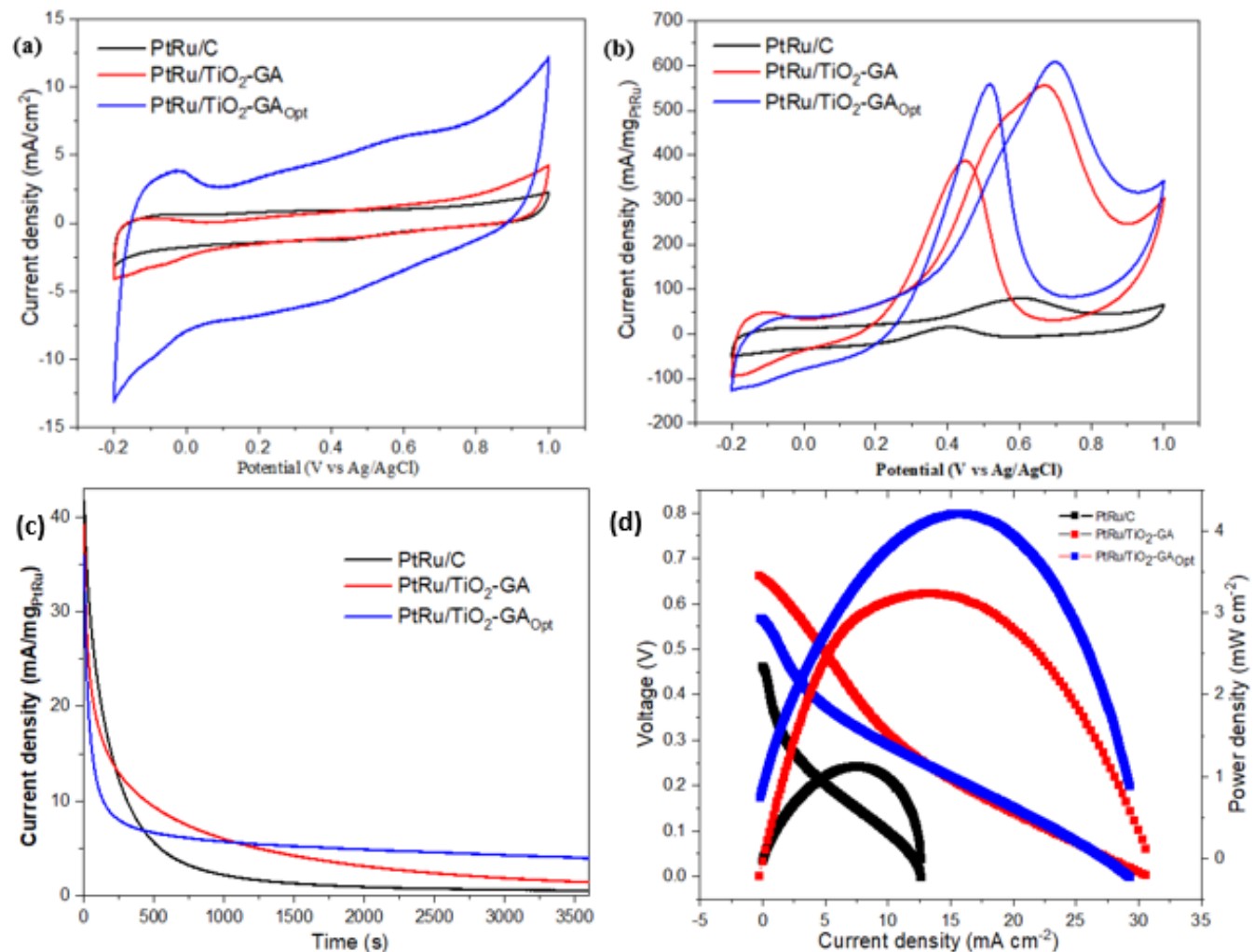

**Figure 11.** Represents the (**a**) CV scan of PtRu catalyst in 0.5 M $H_2SO_4$ electrolyte at 50 mV/s; (**b**) CV scan in 0.5 M $H_2SO_4$ and 2.0M $CH_3OH$ aqueous at 50 mV/s; (**c**) chronoamperometry curve carried out in 0.5 M $H_2SO_4$ + 2.0 M $CH_3OH$ for commercial PtRu/C and different electrocatalysts at 0.4 V vs. Ag/AgCl; (**d**) current–voltage curve for electrocatalyst in 2 M methanol at room temperature.

**Table 6.** Evaluation of the current density results with the different catalyst supports.

| Electrocatalysts | Peak Potential (V vs. Ag/AgCl) | Onset Potential (V vs. Ag/AgCl) | Peak Current Density (mA mg$^{-1}$) | ECSA (m$^2$ g$^{-1}$) | CO Tolerance, $I_f/I_b$ |
|---|---|---|---|---|---|
| PtRu/C | 0.613 | 0.419 | 79.93 | 3.25 | 5.31 |
| PtRu/TiO$_2$-GA | 0.676 | 0.336 | 554.67 | 23.43 | 1.44 |
| PtRu/TiO$_2$-GA$_{Opt}$ | 0.693 | 0.346 | 568.15 | 30.63 | 1.09 |

Based on the equation, Q is the charge density or area under the graph ((C) of the experimental CV) $\Gamma$ (2.1 $Cm_{Pt}^{-2}$) is the constant for the charge required to reduce the proton monolayer on the Pt, and $W_{Pt}$ is the Pt loading ($g_{Pt}$) on the electrode. According to the ECSA calculations, the synthesised electrocatalyst, PtRu/TiO2-GA$_{Opt}$, has the highest value of 30.63 m$^2$g$^{-1}$, compared to PtRu/C (3.25 m$^2$g$^{-1}$). This occurred due to the PtRu crystallite size, as shown in Table 5 from XRD analysis; the PtRu crystallite size for PtRu/TiO$_2$-GA$_{Opt}$ is the smallest and has a high ECSA value. The catalyst and reaction surface area can be enhanced by using the smallest crystallite size. The crystallite size trend is paralleled by the ECSA value trend for PtRu/TiO$_2$-GA$_{Opt}$ and PtRu/C. The various CV curves in

Figure 11b depict the inverted scan, with the minor oxidation peak appearing between 0.3 and 0.6 V vs. Ag/AgCl. During the initial oxidation peak, modest oxidation on the reversed scan, also known as the reversed oxidation peak, occurred, resulting in the formation of incompletely oxidised carbonaceous species. According to the CV data in Table 6, if the value of ($I_f/I_b$) can be formulated, the optimised electrocatalyst combination is lower than the commercial. This is due to particle aggregation. However, the catalytic activity of PtRu/TiO$_2$-GA$_{Opt}$ was 7.6 times that of PtRu/C. This discovery suggests that using an aerogel structure and a metal oxide combination in the electrocatalyst has a high potential for usage in DMFC applications.

As shown in Figure 11b, the electrocatalytic performance of the synthesised electrocatalyst was evaluated using CV. At room temperature, the CV curves for the electrocatalysts PtRu/TiO$_2$-GA, PtRu/TiO2-GA$_{Opt}$, and PtRu/C are measured in 2 M methanol with 0.5 M H$_2$SO$_4$ and saturated N$_2$ gas. The various curves are measured between −0.2 and 1.0 V vs. Ag/AgCl. The peak current density was PtRu/TiO$_2$-GA$_{Opt}$ > PtRu/TiO$_2$-GA > PtRu/C in decreasing order. In comparison to Ag/AgCl, the peak current density of PtRu/TiO$_2$-GAOpt for the MOR appeared to be 0.639 V. Table 6 shows the peak current density and other CV values for all of the samples. PtRu/TiO$_2$-GA$_{Opt}$ has a current density of 608.17 mA mg$^{-1}$, which is 1.1 and 7.6 times greater than that of PtRu/TiO$_2$-GA and commercial electrocatalyst PtRu/C, respectively. This is due to TiO$_2$-GA's unusual 3D structures, which allow for a quick ion/charge transfer channel [66]. This one-of-a-kind characteristic benefits surface chemical processes and increases the activity of the electrocatalyst. Furthermore, the TiO$_2$-GA structure enables Pt and Ru nanoparticles to attach more firmly to the TiO$_2$-GA surface, as illustrated in the surface morphology section in Supplementary Figures S2 and S3.

The superior MOR performance of PtRu/TiO$_2$-GA$_{Opt}$ can be attributed to the XRD discovery that the lowest particle size in the electrocatalyst yields high ECSA values, as proven by CV analysis. Furthermore, the FESEM analysis revealed that the distribution of TiO$_2$-GA$_{Opt}$ support particles was more uniform than the others. Furthermore, the BET study demonstrated that the inclusion of TiO$_2$-GA$_{Opt}$ support, which has good electrical conductivity, a unique network structure, and a high surface area, increases electrocatalyst area and activity. Therefore, this electrocatalyst combination is promising for future usage as a catalyst for methanol electrooxidation in DMFC applications.

The CA results would be used to conduct studies on electrocatalyst durability and stability. Figure 11c compares the anodic peaks in terms of the mass activity of PtRu/TiO$_2$-GA and PtRu/C commercial electrocatalysts towards MOR in 0.5 M sulfuric acid and 2.0 M methanol at a scan rate of 50 mV s$^{-1}$. The electrocatalysts all displayed a quick and dramatic reduction in current density before slowly becoming horizontal during the stability test using CA measurement. The retention value (percent) (Table 7) in the pattern rose after 3600 min, such as PtRu/TiO$_2$-GA$_{Opt}$ < PtRu/TiO$_2$-GA < PtRu/C. The lowering current density ratio of the PtRu/TiO$_2$-GA$_{Opt}$ electrocatalyst was somewhat greater than that of the PtRu/C, however, and this electrocatalyst obtained the maximum current density of all the electrocatalysts in Table 6. This was attributable to the catalyst support's superior dispersion as well as the enhanced utilisation of catalysis [51,67,68].

**Table 7.** Retention rates of electrocatalysts.

| Electrocatalyst | $j_i$ (mA cm$^{-2}$) | $J_f$ (mA cm$^{-2}$) | Retention Rates (%) |
|---|---|---|---|
| PtRu/C | 41.7 | 0.55 | 98.7 |
| PtRu/TiO$_2$-GA | 38.8 | 1.53 | 96.1 |
| PtRu/TiO$_2$-GA$_{Opt}$ | 35.9 | 4.05 | 88.7 |

*2.5. Passive Single Cell Performance*

Based on the single-cell findings presented in the figure, PtRu/TiO$_2$-GA after optimisation had a positive effect on the single-cell performance test of direct methanol fuel

cells. As a result, considerable alterations in methanol oxidation occur. $PtRu/TiO_2$-GA evaluated the single-cell performance of passive mode DMFC and was published for the first time. The current-voltage (I-V) polarisation curves for 22% $PtRu/TiO_2$-GA and 22% PtRu/C electrocatalysts tested using a single DMFC cell at room temperature, as well as 20% $PtRu/TiO_2$-GA electrocatalysts before optimisation are shown in Figure 11d, along with the 3.2 mW $cm^{-2}$ achievements. These molecules could obstruct the surface location of the catalyst where methanol would react, resulting in decreased performance. Since the methanol flow rate cannot be controlled in passive mode, the tank is easily contaminated with methanol intermediate products, which can block GDL and impair DMFC performance. Table 8 compares the performance of the $PtRu/TiO_2$-GA electrocatalyst current density with prior studies in passive mode DMFC. In contrast to before optimisation, the $PtRu/TiO_2$-GA catalyst's optimum conditions demonstrated a considerable improvement in DMFC performance. Once all of the results were compared, the $PtRu/TiO_2$-GA electrocatalyst was found to be comparable to electrocatalysts previously reported by other researchers. The $PtRu/TiO_2$-GA electrocatalyst has significant potential and could be used as an anode catalyst for passive-mode DMFC in the future.

**Table 8.** Comparison of the single cell performance results.

| Study | Electrocatalyst | Power Density (mW/cm$^2$) |
|---|---|---|
| This study | $PtRu/TiO_2$-GAOpt | 4.2 |
| This study | $PtRu/TiO_2$-GA | 3.2 |
| This study | PtRu/C | 1.1 |
| Abdullah et al. [69] | PtRu/TiO2-CNF | 3.8 |
| Ramli et al. [64] | PtRu/CNC | 3.35 |
| Shimizu et al. [70] | PtRu/C | 3.0 |
| Hashim et al. [71] | PtRu/C | 3.3 |

## 3. Experimental Section

### 3.1. Materials and Chemicals

The Pt precursor, $H_2PtCl_6$, with 20% content, was kindly supplied by Merck, Darmstadt, Germany, while the Ru precursor, $RuCl_3$ (45–55% content), supplied by Sigma-Aldrich Co., St. Louis, MO, USA, was used in the synthesis catalyst. Ethylene glycol (EG), Nafion solution, isopropyl alcohol, ethanol, and methanol were obtained from Sigma-Aldrich. Titanium isopropoxide (TiPP, 97%) was obtained from Sigma-Aldrich Co. Graphene oxide was obtained from GO Advanced Solution Sdn. Bhd (Malaysia).

### 3.2. Preparation of the $PtRu/TiO_2$-GA Electrocatalyst

Fabrication of a $TiO_2$-GA composite by a combination of hydrothermal and freeze-dried methods that utilised moderate ultrasonication for 2 h, 50 mg of $TiO_2$ was homogeneously dispersed into 100 mL of GO solution (2 mg/mL). The dilution was then placed in a Teflon-lined 100 mL autoclave and kept at 200 °C for 12 h. To make a $TiO_2$/GA composite, the synthesised hydrogel was carefully washed with deionised water and freeze-dried for 24 h. The electrocatalyst fabrication was carried out using the standard atomic ratio of 1:1 of PtRu catalyst with 20 wt% loaded to $TiO_2$-GA support, as shown in Figure 12. The formation started with the weighing of chloroplatinic acid (Pt source) and ruthenium chloride (Ru source) precursors, followed by blending with sonicated ethylene glycol solutions for 15 min. $TiO_2$-GA was added to the precursor solution after it had been thoroughly mixed, and it was stirred for about 30 min while the pH solution (1 M NaOH) was used to modify the solution to 10. To ensure that the reduction process for this formation was completed, the microwave reduction approach was designed for one minute and then off for one minute, repeated two times. Eventually, the composite was washed and filtered several times with DI water and ethanol and then dried in an oven for 3 h at 120 °C.

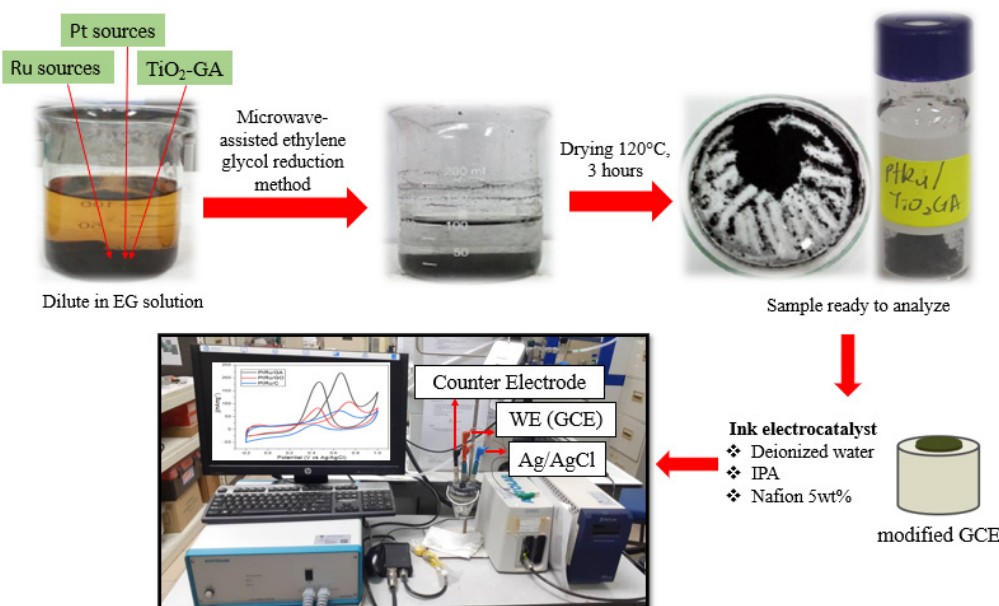

**Figure 12.** Preparation of PtRu/TiO$_2$-GA electrocatalyst.

### 3.3. Electrocatalyst Characterisation and Electrochemical Testing

The crystallinity and presence of the metal alloy in the electrocatalyst samples were characterised using X-ray diffraction (XRD) (D8 Advance, Bruker, Bremen, Germany)). XRD was used to provide the particle size distribution. The pore characteristics were measured using a Brunauer-Emmett-Teller (BET) analysis on a Micromeritics ASAP 2020 instrument (Beijing, China) at 77 K under nitrogen gas flow. A FESEM analysis with a SUPRA 55 VP (Zeiss, Birmingham, UK) was used to examine the surface morphology of Pt-based bimetallic electrocatalyst-supported TiO$_2$-GAs. In addition, energy-dispersive X-ray (EDX) and mapping were performed on all prepared electrocatalysts to estimate the distribution of the electrocatalyst element in samples. Transmission electron microscopy (Tecnai G2 F20 X-Twin, FEI, Hillsboro, USA) was used to measure the catalyst particle size.

For this research, CV and CA electrochemical tests were conducted on the samples and carried out in an electrolyte containing 0.5 M H$_2$SO$_4$ and 2.0 M CH$_3$OH as the fuel solution. At the moment, a three-electrode system was being used at room temperature, with a working electrode consisting of a glassy carbon electrode (3 mm Ø), a counter electrode containing platinum, and Ag/AgCl electrodes serving as reference electrodes. To make preparations for the electrocatalyst ink for the working electrode, 50 μL (5 wt%) Nafion solution, 150 μL DI water, and 150 μL IPA are sorted with 2.5 mg electrocatalyst. Then, 2.5 μL electrocatalyst ink was carefully dropped onto a glassy carbon electrode and allowed to dry. The Autolab electrochemical workstation was used to measure CV with a scan rate of 50 mVs$^{-1}$ and potentials ranging from −0.2 to 1.0 V vs. Ag/AgCl.

### 3.4. Experimental Design
#### 3.4.1. One-Factor-At-A-Time (OFAT)

OFAT is among the approaches that change each parameter at a time while keeping the others constant [72]. Before final optimisation using the RSM method, the OFAT concept is used to reach a target of required factor values. This study focuses on three parameters, the most important of which is the electrocatalyst variable of the temperature composite, TiO$_2$-GA, manufactured with the catalyst. A second parameter investigated is the ratio of Pt to Ru, and the last parameter is catalyst composition for optimised half-cell performance. The variables were selected for this research because they have a robust impact on the results of electrocatalyst elements of DMFC, which is consistent with the findings of other researchers [32,33,51–54]. Depending on metal oxide and/or carbon nanostructure

electrocatalysts supported by PtRu, the values for the variables were chosen as guidance for screening levels via other researchers [22,32,51–54]. The temperature composite, Pt/Ru ratio, and catalyst composition are ranged in this analysis in the ranges of 180–220 °C, 0.5–2.0, and 10–30 wt%.

### 3.4.2. Response Surface Methodology

The central composite design (CCD) was preferred to evaluate the influence of various factors on a specific response. The three factors investigated in this study were the temperature of the composition $TiO_2$-GA, the ratio of Pt to Ru, and the catalyst composition, whereas half-cell DMFC performance was measured using a current density. Design Expert 8.0.6 was used to carry out this method (Stat-Ease Inc., Minneapolis, MN, USA). The factors were graded on a five-point scale ($-\alpha$, $-1$, 0, +1, +$\alpha$). Three factors generate 20 experimental studies with CCD, of which five are performed at a face centre. The second-order polynomial model agrees with this assessment [37]:

$$Y = \beta 0 + \beta 1A + \beta 2B + \beta 12AB + \beta 11A2 + \beta 22B2 \tag{6}$$

where Y is the dependent variable (current density), A and B are the limit values of the independent variable, $\beta$o, $\beta_1$, $\beta_2$, and $\beta_{11}$ are constant coefficients, o are quadratic constants, $\beta_1$ and $\beta_2$ are coefficients for linear terms, $\beta_{11}$ and $\beta_{12}$ are the coefficients for quadratic terms, and $\beta_{22}$ is the coefficient for the second-order interaction terms. The coefficient of regression, analysis of variance (ANOVA), F-value, and *p*-value were used to analyse the constructed regression model. The determination coefficient, $R^2$, measures the fit effectiveness of the polynomial equation model. The optimal values for these three variables were then evaluated by running validation experiments and comparing the values projected by the model. Thus, every test was performed in triplicate to produce an average value.

### 3.5. Preparation of MEA

The anode, membrane, and cathode are three important parts in the production of the membrane electrode assembly (MEA). The membrane Nafion 117 is chosen, and it is treated with hydrogen peroxide ($H_2O_2$) and DI water as shown in a study by Hasran et al. [73] to remove impurities. Besides, the electrocatalyst anode layer was PtRu/$TiO_2$-GA, and the cathode layer was Pt/C based on water-impermeable porous carbon paper. Thin plastic blades helped load the carbon slurry (HiSPEC 1000 carbon powder and 9% PTFE solution) onto the carbon paper surface. After that, the coated carbon layer was allowed to cool before being sintered in a furnace at 350 °C for 60 min to allow a diffusion layer to form at a load of 2 mg cm$^{-2}$. The catalyst is then properly dispersed on the coated carbon support layer using PtRu/$TiO_2$-GA catalyst ink on the anode side and Pt/C on the cathode side, respectively. In this case, carbon cloth is used to prepare the catalyst layer, which is rubbed with carbon slurry and dried in an oven at 100 °C for 1 h. Hence, DI water (64 mg), IPA (96 mg), and Nafion dispersion (170 mg) with an 8 mg cm$^{-2}$ loading were added. In the homogenizer, the solution is dispersed and cast onto the carbon cloth. The anode and cathode are dried at 100 °C for 1 h in the oven. According to the suitability of the research guided by Zainoodin et al. [74], the method of preparing MEA was modified. Finally, clamping the anode, PtRu, and cathode, Pt/C, in the middle is a commercial MEA (Nafion 117 from DuPont) prepared using a heater at 135 °C and a pressure of 15 kg cm$^{-2}$ for 3 min.

### 3.6. DMFC Performance Test

In this study, the DMFC performance testing of PtRu/$TiO_2$-GA before and after optimism was studied and compared with commercial PtRu/C via a polarisation graph using a potentiostat or galvanostat while an MEA with a 4 cm$^{-2}$ active area was installed. The anode was then filled with 10.0 mL of 2.0 M methanol as fuel and tested. PtRu/$TiO_2$-

GA single-cell experiments were carried out at room temperature in passive conditions with PtRu/C.

**4. Conclusions**

In conclusion, the improved PtRu/TiO$_2$-GA electrocatalyst outperformed PtRu/TiO$_2$-GA and PtRu/C in terms of fuel cell performance, which was the major goal of this study. This research is significant since no previous study on adding modified PtRu/TiO$_2$-GA electrocatalysts for DMFC applications has been published. The input factors were chosen as the temperature of the composite (°C), the ratio of Pt to Ru, and the catalyst composition employed, and the output variables were chosen as the current density. Second-order quadratic models were obtained by CCD optimisation in RSM. These models showed a strong connection between projected and experimental outcomes. The best-optimised PtRu/TiO$_2$-GA electrocatalyst was manufactured at a TiO$_2$-GA temperature of 202 °C, a Pt/Ru ratio of 1.1:1, and a catalyst composition of 22 wt%. The PtRu/TiO$_2$-GA electrocatalyst had the best physicochemical performance, with temperatures of 200 °C, 1:1, and 22 wt% for input variables, respectively. The validation test with optimum factors yields a current density of 568.15 mA/mg$_{PtRu}$, with just a 0.6% deviation from the predicted value (564.87 mA/mg$_{PtRu}$). PtRu/TiO$_2$-GA had a maximum power density of 4.2 mW cm$^{-2}$ in the passive single DMFC test, which was 3.5 times greater than PtRu/C. The comparison of electrocatalyst performance and passive DMFC single-cell performance of the current work and other recent research is summarised in Tables 4 and 8. As a result, it was demonstrated that by optimising the PtRu/TiO$_2$-GA electrocatalyst, good performance may be attained to replace PtRu/C in DMFC applications. As a result, the unique combination of PtRu and TiO$_2$-GA 3D materials demonstrated notable activity and stability. Furthermore, this research approach is regarded as a practical, quick, and low-cost way of producing graphene aerogel-based hybrid nano-electrocatalysts, and this form of the catalyst is promising for low-temperature fuel cell applications.

**Supplementary Materials:** The following supporting information can be downloaded at: https://www.mdpi.com/article/10.3390/catal13061001/s1, Figure S1: Activity of the catalyst during the CV electrochemical test; Figure S2. Surface morphology for (a) FESEM of TiO2-GA, (b) FESEM of PtRu/TiO2-GA, (c) EDX analysis of PtRu/TiO2-GA; Figure S3. TEM images of the prepared (a) PtRu/TiO$_2$-GA and (b) PtRu/TiO$_2$-GA$_{Opt}$.

**Author Contributions:** Formal analysis, S.H.O.; Investigation, S.H.O.; Writing—original draft, S.H.O.; Writing—review & editing, S.K.K.; Supervision, S.K.K., S.B. and N.A.K.; Project administration, S.K.K.; Funding acquisition, S.K.K. All authors have read and agreed to the published version of the manuscript.

**Funding:** The study received financial support from the Universiti Kebangsaan Malaysia: DIP-021-028.

**Data Availability Statement:** The data will not be shared due to privacy and confidentiality for the purpose of patent filling.

**Acknowledgments:** The authors gratefully acknowledge the financial support given for this work by the Universiti Kebangsaan Malaysia under: DIP-021-028.

**Conflicts of Interest:** We confirm that the work described has not been published before, it is not under consideration for publication anywhere else, and publication has been approved by all co-authors and the responsible authorities at the institute(s) where the work was carried out. The authors declare that they have no competing interests.

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
