# Peer review of "Anodic Catalyst Support via Titanium Dioxide-Graphene Aerogel (TiO2-GA) for A Direct Methanol Fuel Cell: Response Surface Approach"

_catalysts, doi:10.3390/catal13061001_

Round 1
Reviewer 1 Report
The paper entitled “Anodic Catalyst Support Via Titanium Dioxide-Graphene Aerogel (TiO2-GA) For A Direct Methanol Fuel Cell: Response Surface Approach” synthesized titanium dioxide-graphene aerogel as anode catalyst for methanol oxidation in direct methanol fuel cells The results were reasonably analyzed and discussed. However, the present version can not be accepted. Comments and questions are as follows:
1. The novelty should be further emphasized in the Introduction. There are many literature that reported the TiO2 and graphene supported Pt and Ru as catalyst for methanol oxidation.
2. English should be further improved, and please provide the full name of abbreviation at their first appearance.
3. It is suggested to provide the detailed information about DMFC performance test, including the architecture of fuel cell, the measurement method etc.
4. In Figure 2, the Scale bar was not clear.
5. To ensure the reliability of data, as least four points were required in Figure 5 and Figure 6. In addition, error bar should be provided to verify the reproducibility of results.
6. It is suggested that the standard card and references should be provided in XRD analysis.
7. Authors mentioned that “N2 adsorption-desorption isotherms and pore size distribution analyses were performed at a temperature of 77 K to investigate the porous structure and specific surface area of TiO2-GA composites”. However, it seem that the authors only discussed isotherms curves. Please provide the profile of pore size distribution based on was not analyzed N2 adsorption-desorption isotherms and provide discussion and analysis.
8. For the measurement of DMFC performance in Figure 18, the anode and cathode potentials should be provided to verify that the improvement of DMFC performance was from anode rather than cathode.
Reviewer 2 Report
This work reports Anodic Catalyst Support Via Titanium Dioxide-Graphene Aerogel (TiO2-GA) For A Direct Methanol Fuel Cell: Response surface approach. Authors incorporated an experimental setup that generates the best results to evaluate the effectiveness of these variables on electrocatalysis performance in a fuel cell system using TiO2-GA aerogel. Authors claimed that the response surface methodology regulates the best combination of operational parameters, which include a temperature of composite TiO2-GA, ratio Pt to Ru (Pt: Ru), and PtRu catalyst composition (wt %) as factors (input) and current density (output) as a response for the optimization investigation. This study seems interesting; however, some concerns should be addressed before considering the manuscript for publication in catalysts.
1. The synthesis scheme doesn't show the detailed procedure for the precursor's quantity or molar ratios for fabricating PtRu/TiO2-GA electrocatalyst. Authors are suggested to include every step in detail for a better understanding of the protocols and reproducibility of the results
2. Based on the authors proposed mechanism, electron transfer from support to the catalyst facilitates the MOR. I wonder how authors can verify this mechanism. It will be interesting to elaborate with some evidence.
3. SEM mapping in figure 2 should be labeled with corresponding elements (Fig2 d-h)
4. Figures resolution is not up to the standards for publication. Moreover, the labels and tagging should be in a similar format and font style.
5. Manuscript is overloaded with excessive figures. Authors are suggested to merge some figures for better presentation and move additional figures into supplementary information. A total of 18 figures are too many in the main text. For example, figure 2 and figure 14 show SEM analysis is not necessarily important to be kept in the main text, and figs 16, 17 and 18 can be combined into one figure. I suggest 4-5 figures in the main text.
6. Some references are suggested to update the citations in the revised manuscript. ChemElectroChem 4, 3126-3133 (2017). J. Catal. 43, 1459-1472 (2022). Trends Chem. 4, 886-906 (2022).
